# Machine Learning Driven Prediction of Residual Stresses for the Shot Peening Process Using a Finite Element Based Grey-Box Model Approach

Benjamin James Ralph *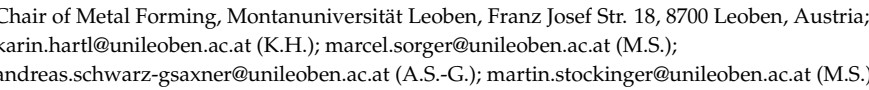, Karin Hartl, Marcel Sorger, Andreas Schwarz-Gsaxner and Martin Stockinger

Chair of Metal Forming, Montanuniversität Leoben, Franz Josef Str. 18, 8700 Leoben, Austria;
karin.hartl@unileoben.ac.at (K.H.); marcel.sorger@unileoben.ac.at (M.S.);
andreas.schwarz-gsaxner@unileoben.ac.at (A.S.-G.); martin.stockinger@unileoben.ac.at (M.S.)
* Correspondence: benjamin.ralph@unileoben.ac.at; Tel.: +43-384-2402-5611

**Abstract:** The shot peening process is a common procedure to enhance fatigue strength on load-bearing components in the metal processing environment. The determination of optimal process parameters is often carried out by costly practical experiments. An efficient method to predict the resulting residual stress profile using different parameters is finite element analysis. However, it is not possible to include all influencing factors of the materials' physical behavior and the process conditions in a reasonable simulation. Therefore, data-driven models in combination with experimental data tend to generate a significant advantage for the accuracy of the resulting process model. For this reason, this paper describes the development of a grey-box model, using a two-dimensional geometry finite element modeling approach. Based on this model, a Python framework was developed, which is capable of predicting residual stresses for common shot peening scenarios. This white-box-based model serves as an initial state for the machine learning technique introduced in this work. The resulting algorithm is able to add input data from practical residual stress experiments by adapting the initial model, resulting in a steady increase of accuracy. To demonstrate the practical usage, a corresponding Graphical User Interface capable of recommending shot peening parameters based on user-required residual stresses was developed.

**Keywords:** python scripting; residual stresses; shot peening; finite element analysis; digitalization; machine learning; smart factory

## 1. Introduction

For the design of dynamically load-bearing components, a certain safety risk is minimized by increasing the service life and improving its estimation. A key aspect in this context is the selected material and its long-term stability under dynamically oscillating loads [1–3]. Numerous machining end contour processes included in the manufacturing of critical components such as milling, turning, or drilling lead to residual tensile residual stresses on the surface. These stresses are counterproductive for the fatigue resistance; therefore, further surface treatment is essential for these components.

There are several mechanical surface treatment technologies available today, pursuing the objectives of implementing residual compressive stresses close to the surface, as well as introducing a work hardened layer. A well-known example is deep rolling, a low-cost method that achieves a comparatively smooth surface, but is limited to elementary, usually rotation-symmetrical geometries [4]. This technique is mainly used for components that require frictionless sliding, where good surface quality is critical for wear. Another alternative is laser shock peening, an efficient method to introduce compressive residual stresses at four times the depth of shot peening [5]. This is achieved by high-energy laser pulses that introduce a shock wave into the material that exceeds the material's yield strength and causes localized deformation. Although this method is gaining popularity, the

investment in such a system is a high-cost proposition. Moreover, the long process times are currently not suitable for an efficient application in production [6]. Additionally, the ball burnishing or roller burnishing method produces a particularly smooth surface [5,7–9]. A related method developed by Lambda Technologies Group is low plasticity burnishing, which is capable to introduce significant residual compressive stresses while initiating comparatively low work hardening. This assists in ensuring permanent compressive stresses when components are used in higher temperature applications. This method has the further advantage that it can be integrated into a variety of machining systems, e.g., CNC lathes [10–14].

Even though there is a strong effort in establishing new and optimizing well-known surface treatment methods, shot peening still is the standard procedure in the manufacturing environment. Irrespective of the mechanical surface treatment chosen, specific knowledge and therefore respective data about suitable process parameters is mandatory to obtain the required results.

To receive a comprehensive data set for the shot peening process, it is mandatory to obtain a significant amount of valid data. This approach requires the execution of an unreasonable amount of practical experiments per workpiece material/sphere material combination. Furthermore, the same amount of upfollowing experiments to receive valid residual stress profiles would have to be carried out. By substituting practical tests with Finite Element Analysis (FEA)-based simulations, this disproportionate effort can be avoided.

The effectiveness of FEA for production processes can be further increased by using state of the art digitalization technologies, taking into account user, processes, and materials [15–17]. One possibility to achieve this objective is the implementation of robust machine learning algorithms. In order to do so, a first decision has to be made regarding the nature of the respective algorithm. In general, three methods are defined: reinforcement learning (RL), unsupervised learning (UL), and supervised learning (SL) [18]. According to more recent work, there are different subordinate algorithms available, which can be used within one or more of these three main techniques [19,20]:

RL: Genetic Algorithms, Simulated Annealing, and Estimated Value Functions;

UL: Decision Tree Analysis (DTA), Rule-Based Learners, Instance-Based Learners, Artificial and Bayesian Neural Networks (NN), as well as Naïve Bayesian Approaches;

SL: Support Vector Machines, DTA, Rule-Based Learners, Instance-Based Learners, Genetic Algorithms, Artificial and Bayesian NN, and Naïve Bayesian Approaches.

For the prediction of residual stresses after the shot peening process, the authors decided to use a SL algorithm, as the nature of this technique is a continuous learning from data provided by an external knowledgeable source. The accuracy of this algorithm depends on internal knowledge about the expected results and, most important, comprehensible input data [19,21,22].

To achieve accurate data sets serving as an input for this kind of simulation, a suitable material model based on reliable material data from practical experiments must be chosen. Therefore, it is essential to implement real-physics-based input variables, which must be obtained under similar conditions as the process to be modeled.

## 2. Fundamentals of the Shot Peening Process and Corresponding FEA

In order to increase the fatigue strength, shot peening is applied as a standard procedure in the production process for structural materials. This method contributes to the service life enhancement of cyclic loaded components [23]. The most notable advantages of shot peening compared to other surface hardening treatments are the good process quality, reproducibility, and applicability to a wide range of materials and component geometries [3]. During the process, the surface of the component is impacted by spheres at high velocities. As a result of the momentum transfer, work hardening is increased directly on the surface which reduces the probability of crack initiation. The plastic deformations induced by the spheres also generate residual compressive stresses in the material to a certain

depth. These stresses are the main inhibitors of crack propagation due to the prevention of crack tip opening and thus increase the fatigue strength. However, this surface treatment does not always contribute to a work piece's service life extension rather than a reduction, as König investigated for Waspalloy in [24]. Although increasing the degree of coverage from the impacting spheres can increase the magnitude of resulting residual stresses, this additional loading for higher strength materials at the surface may contribute to a higher probability of initiating cracks. Therefore, it is crucial to be aware of the influential variables of the process before it is applied in practice. The process itself is variable in numerous aspects, such as the sphere's material and geometry, as well as the impact velocity and the coverage [25,26]. The average sphere radius is about 0.4 mm and they are commonly made of glass, ceramic, cast iron, or steel. A prerequisite for the sphere's material is the higher hardness compared to the shot-peened material. A higher difference between the sphere's and the target's hardness yield higher resulting residual compressive stresses [27]. Additionally, larger sphere radii result in the maximum compressive stresses occurring deeper in the material [28].

In order to achieve the maximum effect on service life extension through this process, these parameters must be optimally adjusted to the material. The maximum achievable residual compressive stresses and the depth of penetration into the material are decisive, since the residual compressive stresses inside the material are balanced by tensile residual stresses in a certain depth. Additionally, the dislocation density introduced by this surface treatment needs to be observed concerning the resulting material behavior. On the one hand, this can prevent the crack initiation [29], on the other hand, it may contribute to the brittleness of certain materials and thus drastically reduce their service life, especially in corrosive environments [30]. To experimentally analyze the residual stresses inside the material, destructive and therefore expensive examinations based on X-ray diffraction (XRD) or using the hole drilling method have to be performed in practice. A time and cost-saving alternative to physical experiments is the numerical simulation, which allows the determination of favorable parameters for the optimal result in advance. In addition, stresses on the surface and in depth of the material can be analyzed to provide a better comprehension of the effectiveness of the process. Several studies have been carried out using FEA to simulate the shot peening treatment. The approaches to simulate this process vary widely in different publications. In [31], Edberg et al. designed a three-dimensional FEA simulation, comparing a visco-plastic strain hardening formulation to a elasto-plastic one analyzing a single shot. This study revealed that the visco-plastic model overestimated the resulting residual stresses by a factor of 1.5. In [32], Majzoobi et al. used a three-dimensional set up applying multiple shot impacts and investigated the shot velocity and coverage effects on the resulting residual stresses. The investigations of Meguid et al. in [33] included the separation distance of the spheres and its impact on the residual stress profile as well as the frictional behavior of AISI 4340. A comparison between the resulting values of an axisymmetric and a three-dimensional numeric model on an aluminum target was conducted by Han et al. in [34] where high emphasis was attached to the interaction of the sphere and the target as well as suitable boundary conditions for the FEA. In [35], Schwarzer et al. investigated the influence of the sphere's impact angle on the resulting residual stresses while Hong et al. focused on the loss of kinetic energy of the spheres as a result of alternating impact angles in [36]. In [37], Mylonas and Labeas addressed a reasonable relation between the quantity of impacts needed in order to receive the results of experimentally obtained residual stress profiles but still reduce computational time. The approach of reducing computational time is also applied in this study by the usage of a two-dimensional setup for the simulation, in order to provide a beneficial tool for the industry, taking into account the results of previous works mentioned in this section.

## 3. Fundamentals and Behavior of EN-AW-6082 T6 under Dynamic Conditions

The material investigated in this study is the age-hardenable EN-AW-6082 aluminum alloy, which is one of the most essential alloying systems for the usage in lightweight

construction due to its balanced properties and good formability. The chemical composition of the used alloy is shown in Table 1.

**Table 1.** Chemical composition of examined aluminum alloy EN-AW-6082.

| Chemical Composition of EN-AW-6082 (wt. %) | | | | | | | |
|---|---|---|---|---|---|---|---|
| Si | Fe | Cu | Mn | Mg | Cr | Zn | Ti |
| 0.87 | 0.42 | 0.08 | 0.57 | 0.66 | 0.02 | 0.2 | 0.02 |

The alloy achieves its strength values primarily through the precipitation of the so-called β-Phase $Mg_2Si$, and further phases such as $AlSi_6Mg_3Fe$ and $Al_{15}(FeMn)_3Si_2$ with suitable ageing after solution heat treatment. Since particularly Mn particles increase the strength of the alloy, while negatively influencing ductility, a homogenization annealing is carried out before forming in practice [38]. The duration of homogenization annealing increases the effect on the reshaping and distribution of particles and therefore reduces the yield stress for extrusion [39]. The highest strength is achieved with the T6 treatment, which consists of a solution heat treatment between 793 K and 813 K for 30 min to one hour in order to dissolute the alloying elements in the matrix. Subsequent quenching creates a supersaturated condition which is immediately followed by the artificial heating treatment, ranging between 423 K and 443 K for 5–20 h, resulting in a peak of precipitation [40–45]. It is common to consider strain-rate sensitivity for the determination of processing parameters and processing maps, as it has a significant impact on fracture behavior [46]. However, the existence of metastable precipitates causes a change in mechanical properties to higher strength values with a reduction in ductility.

EN-AW-6082 also exhibits deficiencies, especially with regard to fatigue resistance under cyclic loading. When used as a component in a chlorine-containing environment such as near industrial production facilities, the corrosion-resistant passive coating cannot withstand the incorporation of chlorine ions in the passive layer. This increases the probability of pitting corrosion. The crack initiation enhanced by this effect leads to a facilitated crack growth under dynamic loading [2]. In order to increase the fatigue strength, shot peening is applied as a standard procedure in the production process for this alloy.

The initial microstructure of the investigated material is shown in Figure 1. The specimen was prepared by electrolytic polishing using the Barker etching method [47]. The microstructure shows a non-textured grain structure with uniform grain size. The emphasis on the age-hardened condition, which is investigated in the present case, is essential in the case of shot peening, since this treatment is applied as a last processing step after heat treatment.

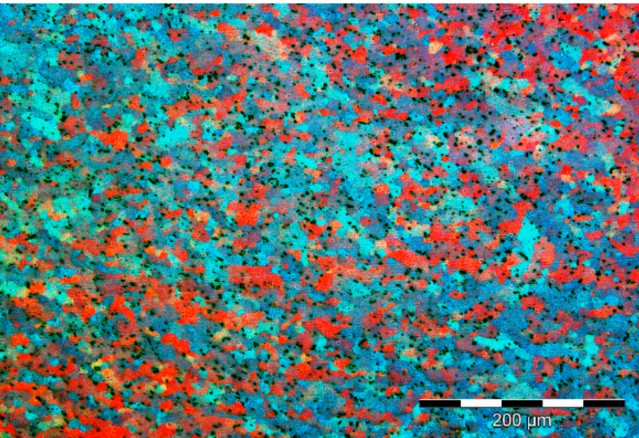

**Figure 1.** Initial microstructure of the EN-AW-6082 specimens investigated.

### 4. The Johnson–Cook Material Model

In order to simulate impact problems such as shot peening, material models are commonly used to represent the material's behavior in the most accurate possible way. Especially for high dynamic impacts, using FEA to model this process is an efficient and effective solution. The most important aspect in this context is the strain rate dependency of a material. Many constitutive models deal with material behavior by dislocation motions and their interactions with lattice defects. For many industrial processing related applications, these models are exceedingly complex and require material data with limited accessibility. Others, such as the Zerilli–Armstrong model, contain a simpler structure, but still include factors that are elaborate to determine, such as initial grain size [48]. In order to provide simplicity and convenience to the user, the Johnson–Cook (JC) material model is establishing itself as the most commonly used material model for impact problems, since it takes both strain rate and thermal softening behavior into account. Nevertheless, it is kept simple, consisting of three terms and five material parameters which are arranged as visualized in (1) [49].

$$\sigma = \left( A + B\varepsilon_p^n \right) \left[ 1 + C \ln\left( \frac{\dot{\varepsilon}_p}{\dot{\varepsilon}_0} \right) \right] \left[ 1 - \left( \frac{T - T_t}{T_m - T_t} \right)^m \right] \tag{1}$$

The first term refers to strain hardening during plastic deformation including the plastic strain $\varepsilon_p$, the yield strength of the quasi-static condition $A$, the strain hardening constant $B$, as well as strain hardening exponent $n$. The second term relates to the material's behavior under different strain rates with the strain rate sensitivity coefficient $C$ as a result of different strain rates $\dot{\varepsilon}_p$ normalized to a quasi-static strain rate $\dot{\varepsilon}_0$. The third term describes the material behavior under temperature influence including the reference temperature $T_t$, the melting temperature $T_m$, and the thermal softening exponent $m$ [49]. The localized strain acquired through the shot peening process is limited, resulting in a small energy input due to the deformation process, even at high strain rates. For this reason, the thermal input due to the plastic deformation of the impinging spheres at the surface is neglected in the JC material model for this framework. Therefore, (1) can be reduced by the third term, resulting in (2).

$$\sigma = \left( A + B\varepsilon_p^n \right) \left[ 1 + C \ln\left( \frac{\dot{\varepsilon}_p}{\dot{\varepsilon}_0} \right) \right] \tag{2}$$

The parameters of the first term can be determined by using (3).

$$\ln(\sigma - A) = n \cdot \ln(B\varepsilon) \tag{3}$$

$A$ can be derived from the initial flow curve under quasi-static conditions. The slope $n$ can be determined graphically by plotting a trend line while $B$ can be expressed by solving the exponential function. The parameter $C$ includes tests for higher strain rates. To receive $C$, (2) has to be arranged as demonstrated in (4).

$$\frac{\sigma}{(A + B\varepsilon^n)} = 1 + C \cdot ln\left( \frac{\dot{\varepsilon}}{\dot{\varepsilon}_0} \right) \tag{4}$$

By plotting the left term of (4) against the logarithmic strain rate ratio, $C$ can be obtained directly from the resulting trend line.

Particular attention is required for the comparison of the determined material parameters with literature values, especially the quasi-static strain rate used ($\dot{\varepsilon}_0$), as this value often varies in a range between $10^{-4}$ and $1\,\text{s}^{-1}$. Another disadvantage regarding literature-based JC parameters is the test setup used to determine these values. For quasi-static stresses, the tensile test is usually selected in literature for the simplicity of the method. For particularly high strain rates, the strain rate sensitivity is frequently determined using the Split-Hopkinson pressure or tensile bar [50]. It should be noted that the stress states

differ in these test methods. The main disadvantage of tensile tests is the instability of the deformation due to geometric deconsolidation processes after the ultimate tensile strength is reached. In contrast, the upsetting test provides steady strain hardening. The critical aspect here, in addition to the frictional conditions at the dies, is the barreling of the specimen. As a result of this phenomenon, the uniaxial load state cannot be ensured [51]. The comparison of the determined material parameters with those from literature revealed deviations in the values. One reason might be that some of the tests performed were carried out under tensile stress conditions. Besides, there might be differences between the chemical compositions of the materials studied. Slight differences in the heat treatment route for the T6 condition could also be responsible for these divergences. For this reason, separate tests should be carried out with the specific material used, in order to eliminate these variations. The different parameters from the literature are listed in Table 2, whereas temperature is not listed due to the lack of definition within the investigated publications. Accordingly, it is essential to arrange the test setup in such a way that it comes closest to real conditions of usage. For the simulation of shot peening processes, the upsetting test is most similar to the compressive stresses introduced by the spheres at the surface. For low degrees of deformation, uniaxial deformation can be also provided, which is why the experiments carried out in this study are based on this principle.

**Table 2.** Material parameters for the JC model for EN-AW-6082 T6 from literature sources.

|  | $A$ [MPa] | $B$ [MPa] | $C$ [-] | $n$ [-] | $m$ [-] | $\dot{\varepsilon}_0$ [s$^{-1}$] |
|---|---|---|---|---|---|---|
| [52] | 250.00 | 243.60 | $7.47 \times 10^3$ | 0.17 | 1.31 | 1.0 |
| [53] | 305.72 | 304.90 | $4.37 \times 10^3$ | 0.68 | - | $10^{-3}$ |
| [50] | 277.33 | 307.93 | $3.2 \times 10^3$ | 0.69 | 1.28 | $10^{-4}$ |

## 5. Experimental Setup

For the determination of the material parameter of the investigated alloy EN-AW-6082 T6, cylindrical samples with a diameter of 8 mm and an initial height of 12 mm were obtained from an extruded rod material. To receive the T6 condition, all specimens were solution-annealed at 803 K for one hour, followed by water quenching. After these steps, age hardening at 443 K for another five hours was carried out. For the determination of realistic material parameters, the specimens were compressed longitudinal to the extrusion direction at room temperature on the Gleeble 3800 thermal-mechanical Simulator, using the Hydrawedge module at constant strain rates of $1\,\text{s}^{-1}$, $10\,\text{s}^{-1}$, and $100\,\text{s}^{-1}$. The Hydrawedge module is especially designed for the simulation of forging and forming processes requiring a high strain rate, as it is capable of significantly reducing ringing of the hydraulic ram. The capability of high-speed deformations allows the generation of flow curves, which are relevant for the shot peening process. As shown within Figure 2, a graphite foil was additionally placed between both contact surfaces to reduce the friction between specimen and anvil, thus ensuring a uniform stress state during compression.

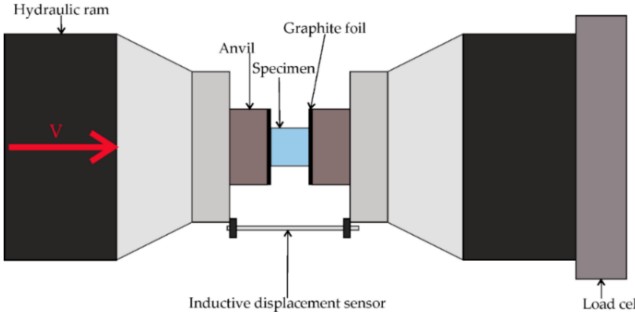

**Figure 2.** Experimental setup for the obtainment of JC material parameters.

Table 3 shows the resulting JC parameters, derived from the practical experiments and calculated according to Section 4. The experiments were carried out until a strain of 0.035 was reached, as higher strains are not relevant considering the shot peening process.

**Table 3.** Material parameters for the JC model for EN-AW-6082 T6 obtained from practical experiments.

| *A* [MPa] | *B* [MPa] | *C* [-] | *n* [-] | *m* [-] | $\dot{\varepsilon}_0$ [s$^{-1}$] |
|---|---|---|---|---|---|
| 385.02 | 116.01 | $7.97 \times 10^3$ | 0.50 | - | 1.0 |

## 6. FEA Setup and Resulting Data Mining Algorithm

For the implementation of the initial state white box model, a fundamental Abaqus input script was defined in first instance. This script contains all necessary input parameters for the simulation model to be automated and is scripted within the Abaqus Python environment. Table 4 shows a brief overview of the most important variables changeable within this input script.

**Table 4.** Variables changeable within the Python input script.

| Input Variable | Functionality |
|---|---|
| Radius | Possible variation in sphere radius |
| x_specimen | Width of investigated specimen |
| y_specimen | Depth of investigated specimen |
| rows | Number of rows of spheres |
| angle | Angle of sphere impact (initially 90°) |
| number_spheres | Number of spheres (per defined rows) |
| delta_x | Horizontal distance between each sphere |
| delta_y | Vertical distance between each sphere |
| row_offset | Offset between different rows |
| step_time_shot | Step time related to the impact phase |
| dens_mat; YM; pois; | Density and elastic behavior of investigated material |
| *A*; *B*; *n*; | JC material parameters for the investigated material |
| *C*; eps_dot_0 | Strain hardening parameters according to the JC model |
| damping_time | Additional step time for stress oscillation analysis |
| friction_coefficient | Defined friction state between specimens and impacting spheres |
| field frames | Number of field output frames within each step |
| v_shot | Shot velocity of spheres |
| mat | Density of spheres (depending on the material) |
| fine_mesh_region | Mesh size of direct impact zone |
| ground_mesh_region | Mesh size of the remaining geometry |
| RS_node | Node set definition for the residual stress analysis |

In order to keep the number of degrees of freedom (dof) for the upstream data analysis reasonable, only the variables v_shot, radius, mat, elastic, and JC parameters of the investigated material (Section 3) were changed. For a further extending of simulation dof, a link between the Python input script and the overlaying automation layer is prepared. The fundamental FEA is defined as dynamically explicit, with widely used element type CPS4R (mesh size 0.01 mm) and a steady friction coefficient of 0.3. To achieve a high shot peening coverage rate on the specimen's surface, 90 spheres within three different rows were created, with a horizontal and vertical distance of 0.025 mm and a vertical offset between each row of 0.02 mm. The specimen's length as well as width was defined with 1.0 mm. Additionally, the impact angle was set to 90° and not changed in this study. To avoid contact definition dependent errors, a loop within the script automatically defined a surface-to-surface contact between each sphere and the target. Table 5 shows the resulting parameters varied within this paper.

**Table 5.** Varied variables within this case study.

| Varied Input Variable | Range (Step) |
|---|---|
| mat | Mat 1.0 (steel spheres)/Mat 2.0 (glass beads) |
| Radius | 0.1–0.5 mm (0.05) |
| v_shot | 30–200 m/s (10) |
| $A$; $B$; $n$; $C$; eps_dot_0 | Literature value (Table 2, [53]) and values obtained (Table 3) |

Figure 3 shows the visualization of an exemplary setup for one defined sphere radius. Depending on the varying radii of the respective spheres, the resulting point mass of each sphere changes. To reduce computational time for the required simulations, the spheres were defined as rigid. For the automated data generation, the Abaqus GUI was excluded from the solver operation.

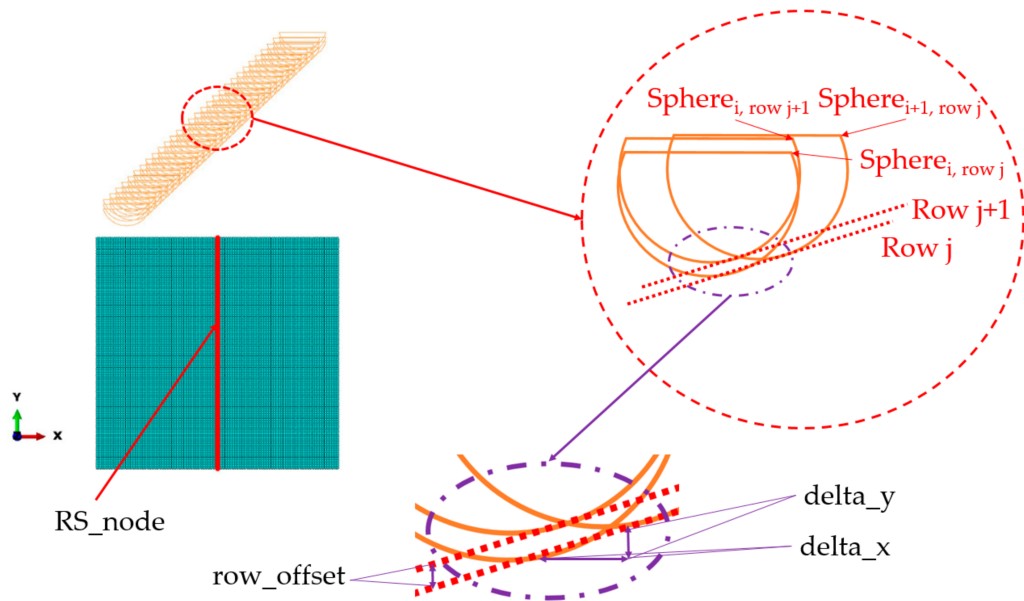

**Figure 3.** Visualization of the experimental setup and definition of geometric variables.

For the development of the white-box model, an initial database with all resulting residual stresses for each node included in the RS_node node set has to be created. This database also includes the different impact velocities and sphere diameters and serves as a basis for the initial GUI. In order to receive the steady-state residual stresses, the resulting amplitude at each respective node within the node set was analyzed. To consider a residual stress value for a node within RS_node as steady, the residual stress amplitude $\Delta\sigma$ for this node at a specific time increment has to be underneath 10 MPa (Figure 4). The fulfillment of this condition is checked within the initial Python algorithm. In this case, for a step time of $10^{-3}$ s, the condition is valid for each node within all performed simulations. The steady-state residual stress was returned and stored in the master database. As a result, one stress value for every 10 μm in each simulation is obtained.

Figure 5 visualizes the programming logic for the creation of this database, starting by the initial input script.

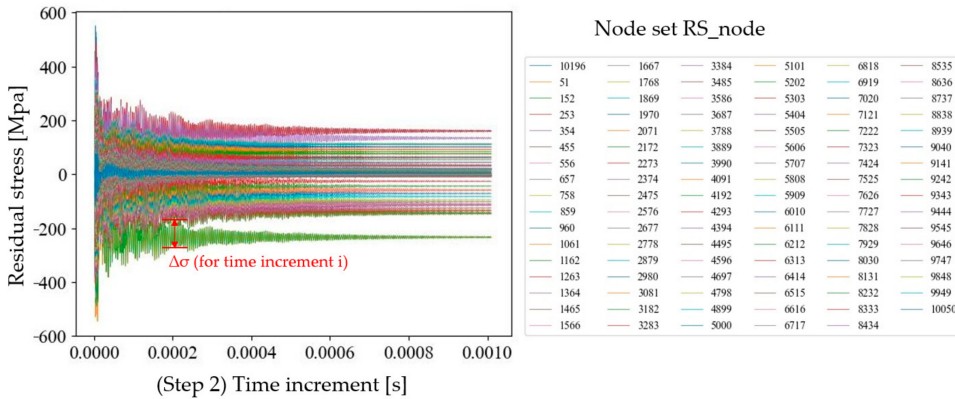

**Figure 4.** Exemplary residual stress amplitudes over step time with included nodes in RS_node.

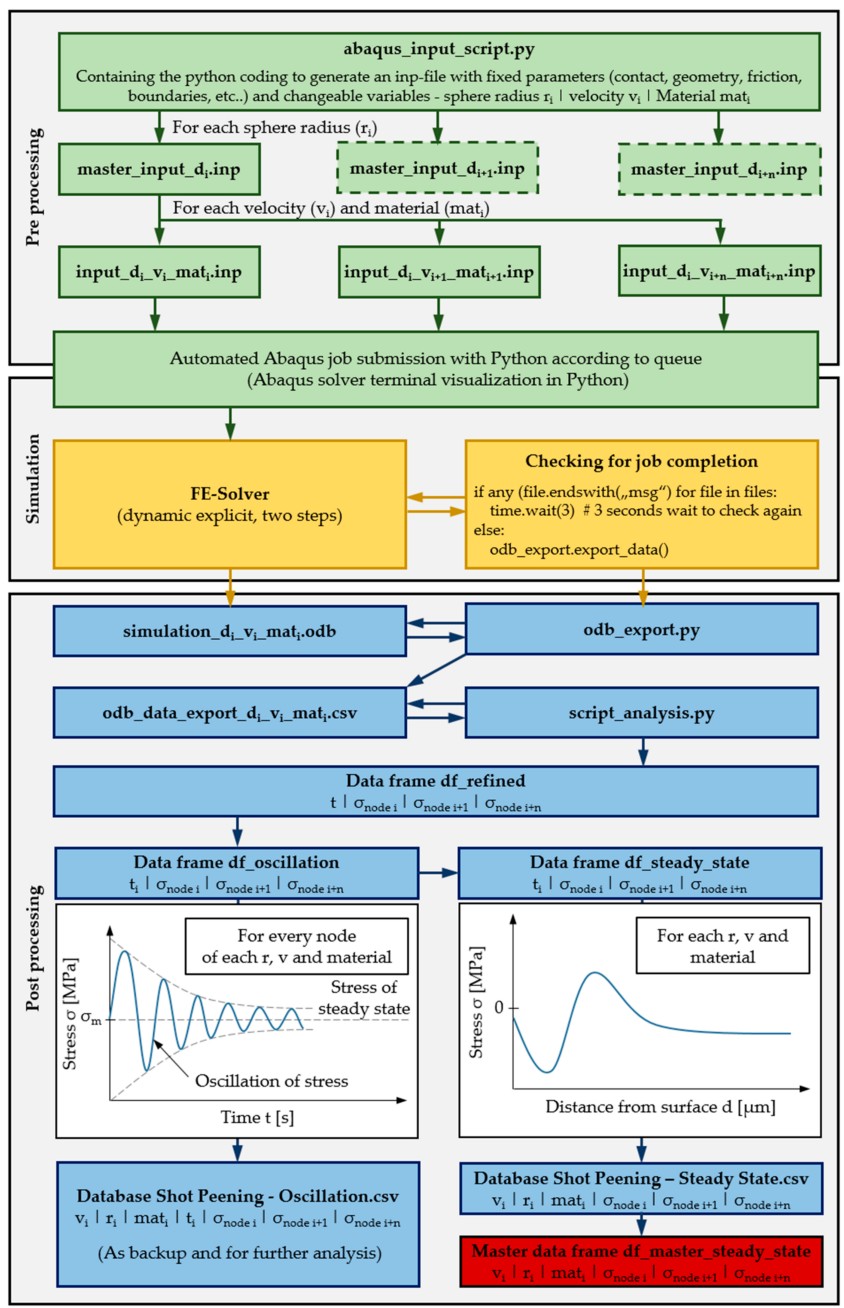

**Figure 5.** Programming logic for the obtainment of the database and master data frame from FEA data.

The master data frame extracted from the steady-state database contains all necessary information for further analysis and implementing the initial white-box-model-based logic. Figure 6 shows the comparison between different velocities for one exemplary sphere diameter (0.4 mm), whereas both investigated sphere materials (steel (red) and glass beads (blue)) are visualized. Additionally, the results for the JC material parameters from [53] are shown (steel spheres (green) and glass beads (orange)).

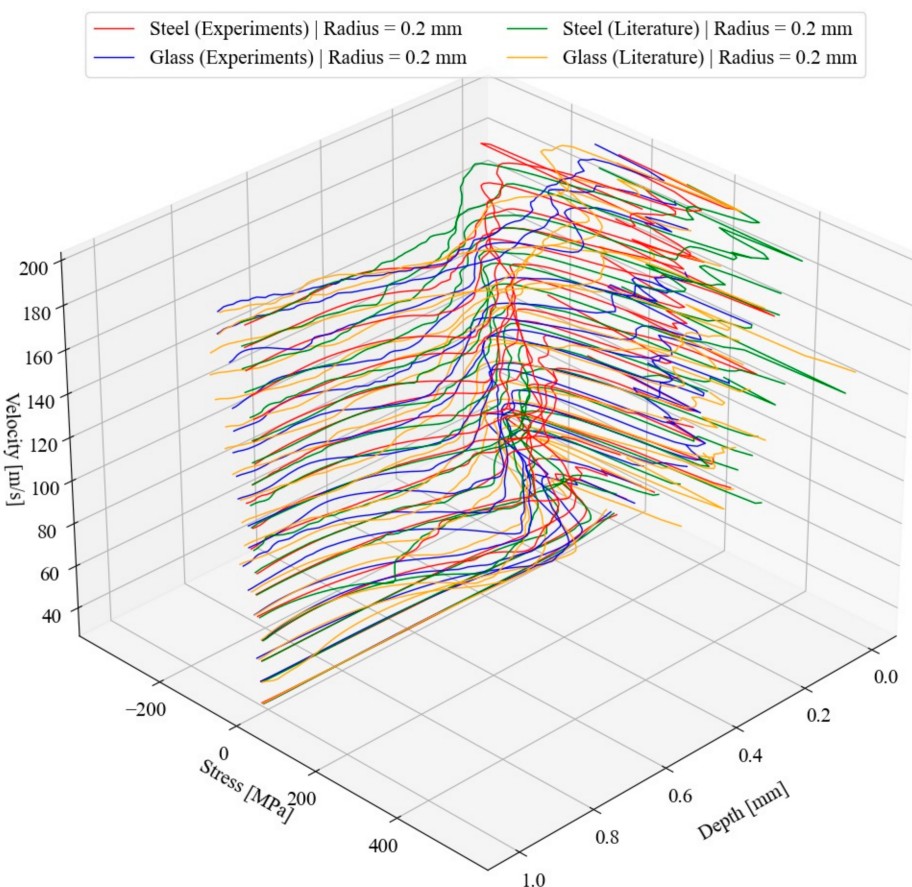

**Figure 6.** Resulting residual stress profile for a defined sphere diameter (0.4 mm) for the JC parameters obtained experimentally (steel spheres (red), glass beads (blue)) and alternative parameters derived from [53] (steel spheres (green), glass beads (orange)).

As demonstrated in Figure 6, a significant difference between the JC parameters determined from literature and own experiments can be seen, for the reasons explained previously in Section 4. In general, the impact of steel spheres results in higher residual stresses within comparable velocities and diameters. This effect can be explained by the higher resulting momentum of the iron-based sphere material, as the density is 3.1 times higher than the density of glass. The observed tensile stresses at the surface are a result of the material flow through adjacent impacts. This effect can be enhanced by the rigid definition of the spheres as well as the chosen mesh size. As the main objective of this framework is to obtain valid residual stress minima under reasonable computational time, this divergence was not considered any further [54].

Figure 7 shows the same sphere material and material parameter variation for a steady velocity (100 m/s) with varying sphere diameters (0.2–1.0 mm). The increase in maximum negative residual stresses with bigger sphere diameter can be explained again by the higher resulting momentum for a steady velocity [28].

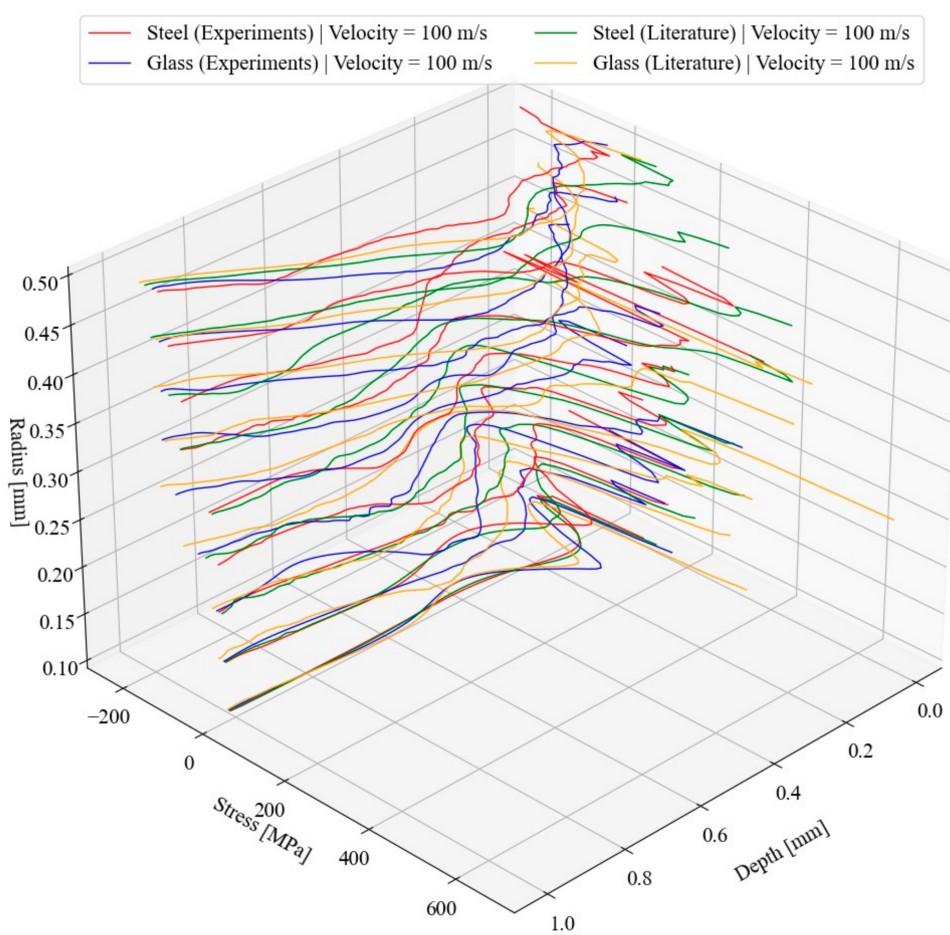

**Figure 7.** Resulting residual stress profiles for a defined velocity (100 m/s) for the same variations defined within Figure 7.

Figure 8 illustrates the difference between literature values and the data obtained from the experiment exemplarily.

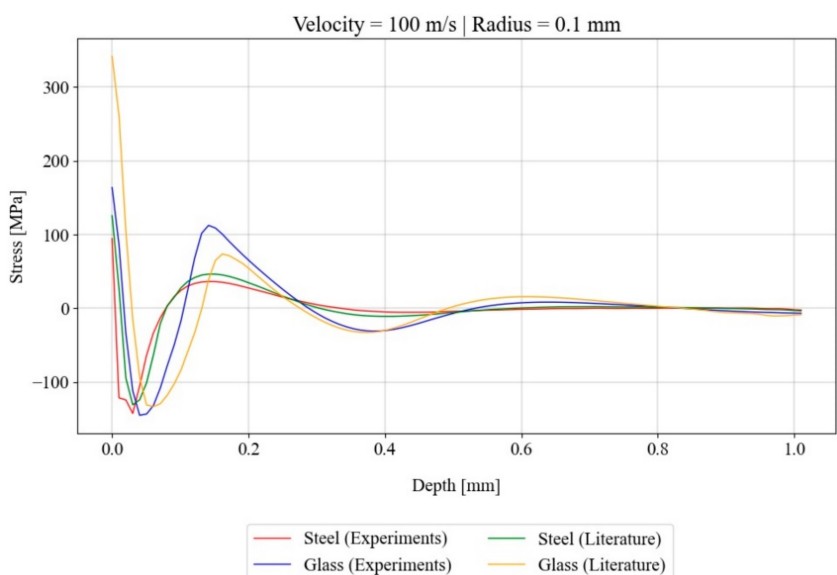

**Figure 8.** Resulting residual stress profiles for velocity = 60 m/s and a sphere diameter of 0.4 mm for literature and experimental data.

## 7. Development of the Initial White-Box Model for the Residual Stress Prediction

Figure 9 visualizes the initial white-box logic, beginning with the input parameters defined by the respective user to the final values returned from the algorithm.

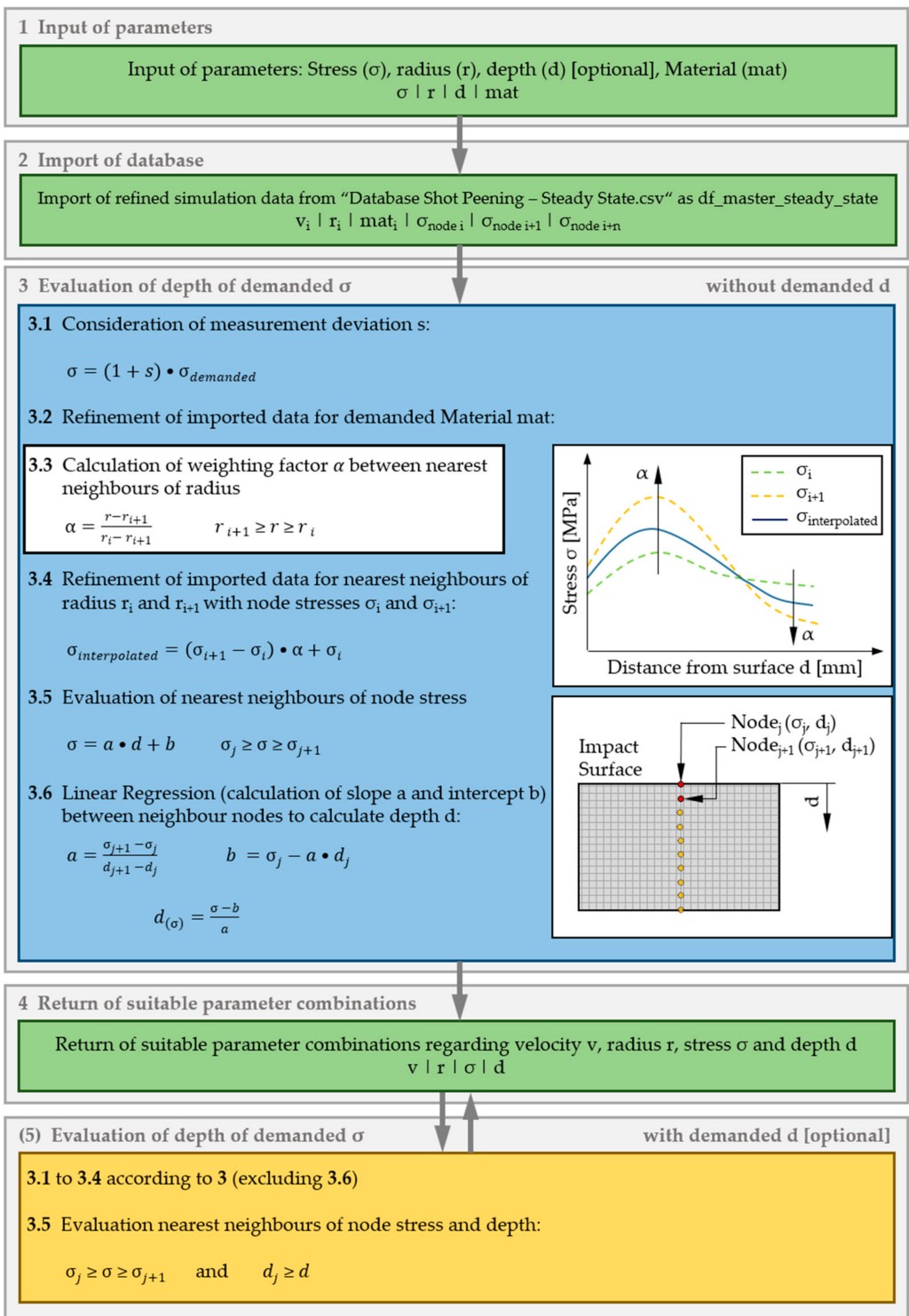

**Figure 9.** Algorithm for the transformation of user input data (real sphere diameter and desired residual stress, optionally required depth) into shot peening parameters (velocity options suitable for the defined input) by using the master data frame defined in Figure 6.

For the user to be able to adapt the initial sphere diameter to the real value, the model has to be capable of interpolating within the given data set. To achieve this, an interpolation scheme, including a linear weighting factor $\alpha$ which interpolates between given boundaries of the initial (FEA-based) data set, was defined. For the practical usage, the respective user is able to define the desired residual stress required for the individual case. Additionally, it is possible to define the desired depth in which the specified stress value should be obtained. If no depth is defined, the user gets a data frame which includes all shot velocities fulfilling the defined input value, including the depth in which the residual stress is reached first. To ensure that the calculated value will be reached in practice, a security factor s was set in the back end, which multiplies the input stress value with the factor 0.2.

## 8. Experimental Data-Driven Machine Learning Algorithm

As within every simulation, a deviation between the calculated results and experimentally determined data occurs. To close this gap in an efficient and sustainable way, the possibility of including actual test data in the model is considered, whereas the actual test data can be gained from different experiments (e.g., XRD measurements). In general, these results contain a few data points for each experiment carried out. To be able to adapt the initial FEA-based data cloud within the master data frame, at least four experiments have to be executed, analyzed, and transferred into the Python environment. These experiments have to be within a defined range of velocities ($\Delta v < = 30$ m/s) and sphere radii ($\Delta r < = 0.2$ mm). Based on this data set, non-linear functions with a sufficient amount of respective supporting grid points (initially 100 per three original data points) are created. For more complicated residual stress profiles, this range must be decreased to ensure accuracy. Based on this additional data, the curves received from the FEA within the range of the experimental data sets are overruled and excluded from the master data frame and steady-state database. Furthermore, interpolations that include experimentally obtained curves change significantly. This procedure is carried out automatically within a Python algorithm, which leads to a steady increase of data-driven analytics. This data is not directly connected to real-physics, which includes black-box approaches within the initially white-box model, resulting in a grey-box model. Figure 10 demonstrates this paradigm change over increasing experimental data infeed.

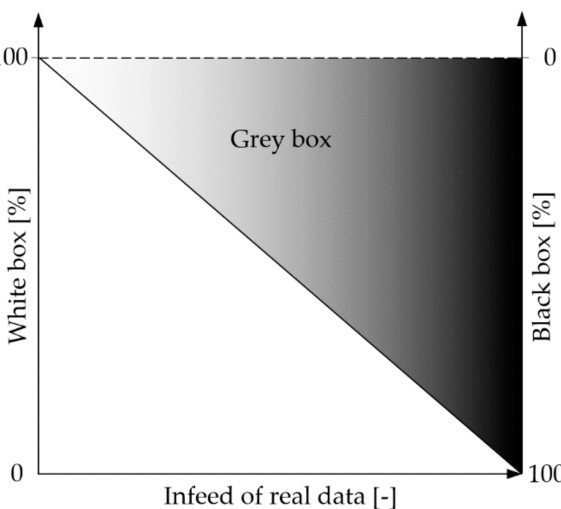

**Figure 10.** Change of model characteristics with increase of infeed data: the original FEA and real-physics-based model is overruled with more data from practical experiments.

Figure 11 shows the logic behind this machine learning approach, programmed within the same Python environment. To smoothen the resulting experimental data points without producing overfitting and therefore unrealistic behavior, a non-linear, second-order fitting approach between experimental data points was chosen. For the same purpose, a mean

value between two overlapping functions for the same data point was used. The resulting second order functions serve as boundaries for the creation of support data points, to be able to interpolate between the new resulting data sets with the same algorithm as for the initial white-box model.

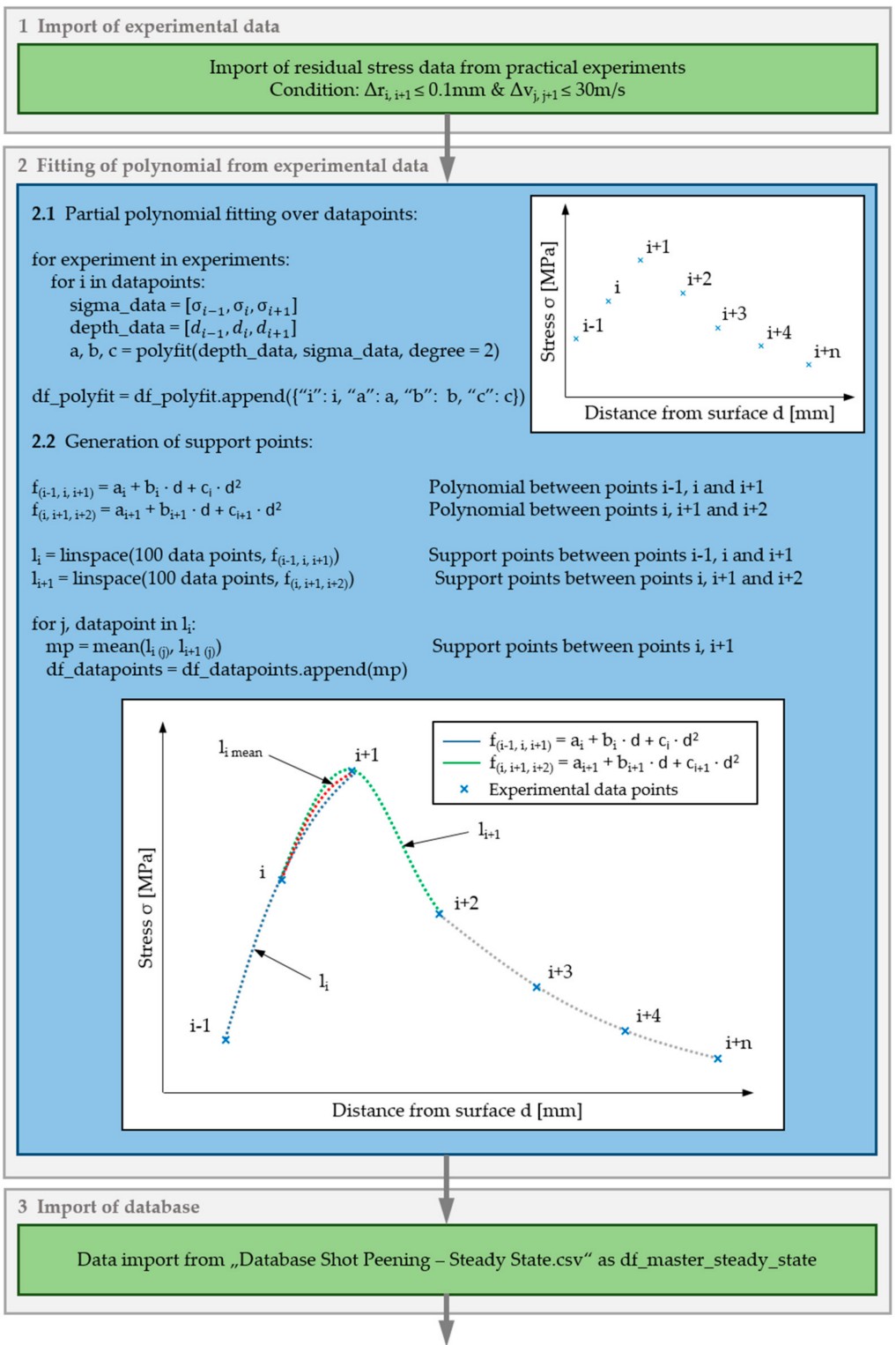

**Figure 11.** *Cont*.

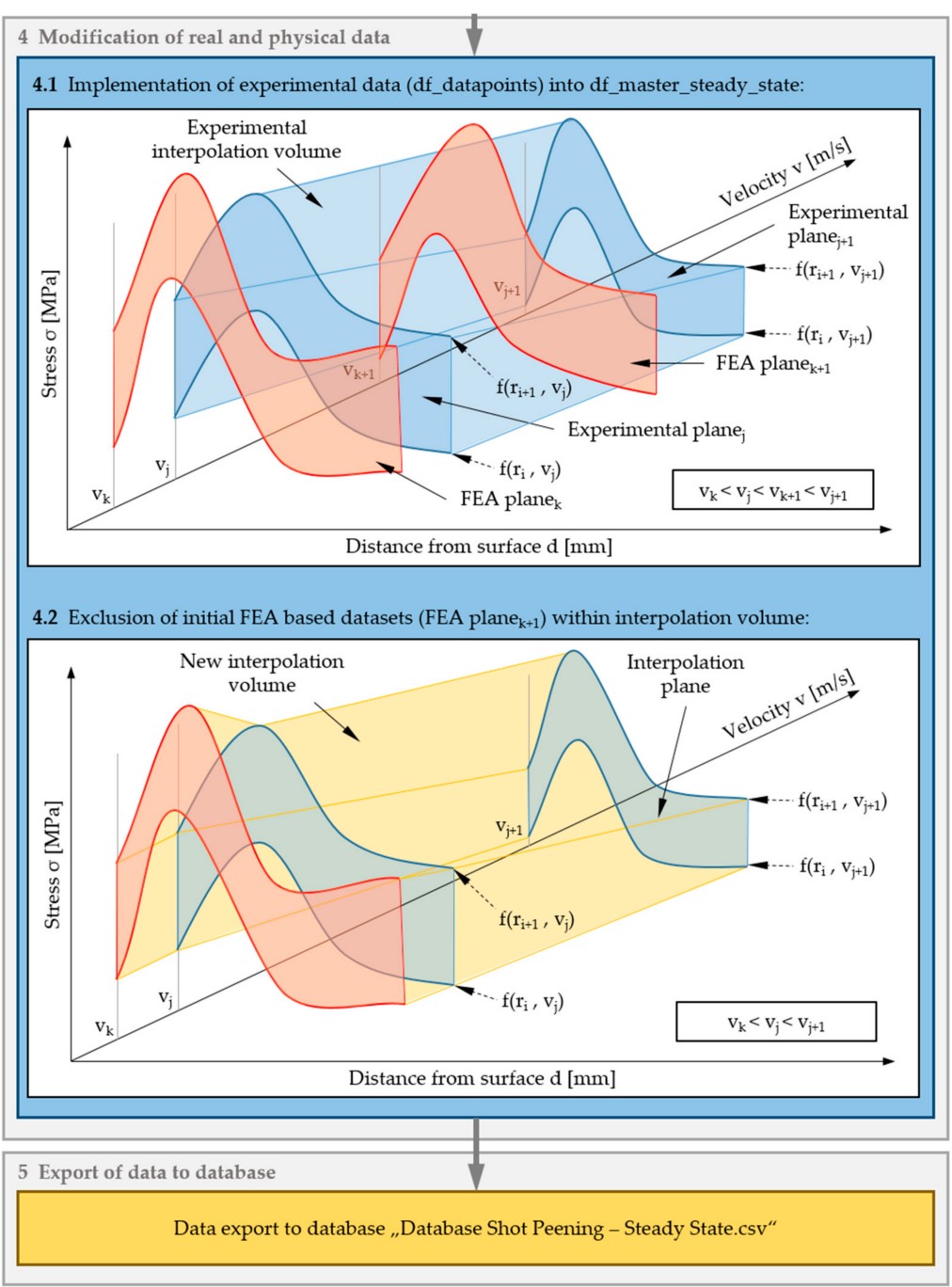

**Figure 11.** Python logic implemented to adapt the initial FEA based white-box model by adding data from residual stress experiments. 1: Import data; 2: generation of support data points from experimental data and storage in a new data frame; 3: loading master data frame; 4: import data points from 2 and overrule data points of the master data frame to increase prediction efficiency; 5: overwrite master database with new data points.

## 9. Graphical User Interface

Based on the logic explained in Sections 7 and 8, a simple and user-friendly GUI was developed, using a C++ based open-source visualization environment. Due to an included library package within the Python environment, a direct programming within the same environment is possible. Figure 12 visualizes the automatic interaction between the resulting GUI and the algorithm developed.

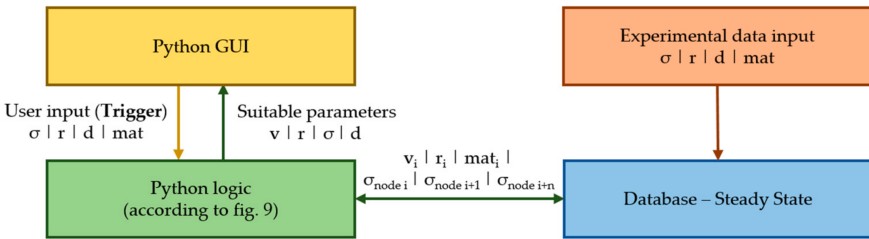

**Figure 12.** Interaction between the developed residual stress algorithm and GUI. To avoid confusion of respective users, the input of experimental data from practical experiments is excluded from this visualization.

Figure 13 shows the implemented GUI without optional definition of desired depth.

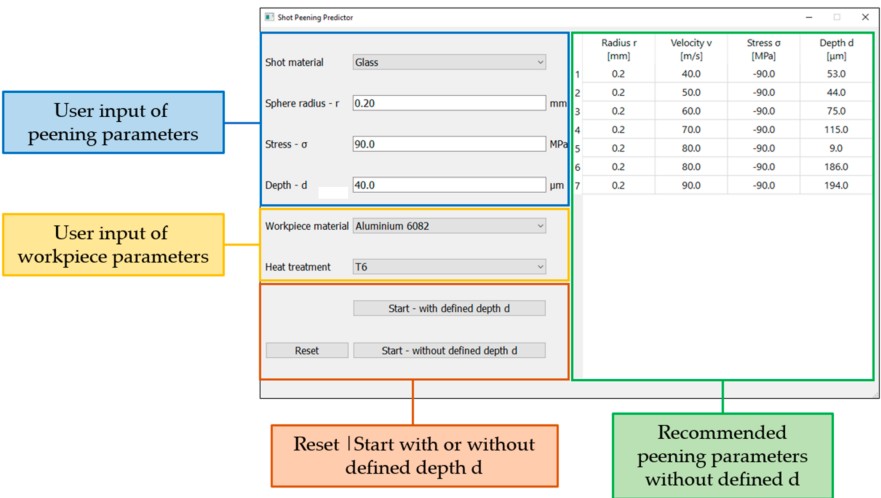

**Figure 13.** GUI with exemplary values for the prediction of residual stresses (without user-defined stress-corresponding depth).

As can be seen in Figure 13, a range of different velocities for the user-required residual stress is returned. If the stress value is necessary within a certain depth, the back-end algorithm changes, resulting in a recommendation for only those shot peening parameters, which result in a smaller depth while fulfilling the required stress (according to Figure 9). Figure 14 demonstrates this by using the same exemplary variables as in Figure 13.

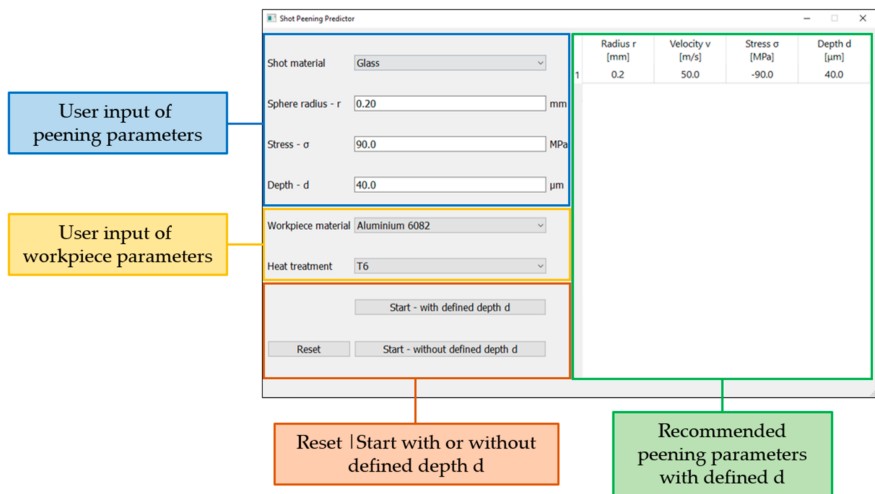

**Figure 14.** GUI with exemplary values for the prediction of residual stresses (with user-defined stress-corresponding depth).

## 10. Results

This paper describes the development of a residual stress prediction module for the shot peening process. In order to demonstrate the logic implemented, an EN-AW 6082 T6 alloy was examined to obtain valid input parameters for the FEA simulation. This FEA model is set up according to literature [26,32,54–56], whereas the reduction of computational time without losing required accuracy was focused on. As a result, over 350 simulations with varying input parameters were automatically executed, resulting in residual stress profiles within common shot peening process ranges for two different sphere materials, 18 different velocities, and ten different sphere sizes. These simulations serve as a basis for the data mining algorithm introduced in Section 7. To enhance the predictor's accuracy, an algorithm for the implementation of experimental data from residual stress tests was additionally implemented. This algorithm is capable of overwriting the initial database within a defined range. For the usage in a production environment and for demonstration to interested parties, a user-friendly, front-end GUI was created, using the same open-source environment as for the logic introduced by the authors.

## 11. Discussion

Due to the ongoing fourth industrial revolution, the technologies implemented in the metal processing and manufacturing environment change significantly. Recent developments in automatic data exchange between production systems do not just increase the productivity within the production operation. The implementation of standardized interfaces additionally offers new possibilities to include other technologies into the process chain with reasonable effort. Numerical simulation, especially FEA, is a common tool in research and development, whereas the direct integration into the process chain is not state-of-the-art in practice. Nevertheless, the possibilities and potential advantages of FEA are pointed out recently in current literature [57,58]. The framework developed by the authors offers the possibility to be implemented into a digitalized production network. The algorithms introduced are programmed completely open-source, which allows interested companies the implementation without high economic barriers. Furthermore, the FEA solver used can be exchanged with every other software package suitable, as long as an interface to an open-source programming language is available. Despite the advantages of the ongoing digitalization and data-driven modeling, real-physics-based engineering has to be included to a certain extent. For the shot peening process, the relationship between workpiece and shot peening material as well as process parameters is complex. Using only black-box approaches would result in an unreasonable amount of required data from practical experiments to be obtained. On the other hand, using only real-physics-driven models often do not consider influences occurring in the manufacturing environment (e.g., sensor offset of respective aggregates, deviations from executed experiments due to different users). The combination of both techniques, although, can reduce the effort as well as deviations, offering an efficient and effective possibility to enhance the production process. Another advantage of the framework introduced in this work is the possibility of extension for all kinds of materials as well as according varieties in heat treatments, as already implemented in the respective GUI. Due to the possibility of changing the interpolation range within the machine learning algorithm, more complex residual stress profiles can be predicted with similar accuracy. However, it is important to note that smaller interpolation ranges result in a higher amount of required input data.

The GUI is designed under special consideration of user-friendliness, giving respective technicians the possibility to choose between two different initial options. Furthermore, the back-end programming carried out in Python ensures fast understanding and can therefore be used for educational purposes. The high connectivity provided within the Python environment allows easy coupling to superordinate networks, enabling users to connect the process simulation easily into a digitalized production system. For this purpose, the two-dimensional setup of the described FEA model should be the optimal compromise between accuracy and efficiency. Nevertheless, for more complex geometry (e.g., bevel,

material steps), a three-dimensional approach is recommended, as the difference between experiments and simulations for more complex geometries cannot be neglected. As the simulation model is based on Python, the implementation of such variations as well as the transformation to a 3-D model can be done shortly. Furthermore, by slightly adapting the initial post-processing, the resulting three-dimensional stress state can be easily obtained.

## 12. Conclusions and Outlook

In this article, a white-box-based framework for the prediction of residual stress profiles after shot peening treatments based on FEA simulations is presented. To include decisive influencing factors, the shot velocity, the sphere's diameter, and the material parameters were varied. According to this framework, a GUI was developed that enables the user in industrial environments to insert preferred residual stresses that should be obtained, receiving the optimal process conditions for this case. Due to the reduction of the simulation setup by using a two-dimensional FEA simulation that is based on the JC material model, the underlying algorithm presents a reasonable fit between efficiency and accuracy. The entries of the JC model can be extended for different materials based on a few practical experiments. The possibility to enhance accuracy of the predictions is given by the ability of the user to insert experimentally investigated resulting stress profiles, which the model adopts while cancelling imprecise entries.

To enhance the usage of the introduced algorithm, additional experiments to obtain valid input parameters from different materials are planned. Based on this additional data, other materials of interest will be inserted into the database. Further results from XRD-based residual stress experiments will also be included for the investigated material as well as additional materials, resulting in a significant increase of accuracy of the algorithm.

The model presented will be implemented within the Smart Forming Lab at the Chair of Metal Forming, connected with different types of Cyber Physical Production Systems by an open-source based MES. The main objective for this specific algorithm is to calculate accurate process parameters for processed workpieces, in order to increase the effectiveness and efficiency of the value chain, from casting to recycling. A possibility to extend this model is the incorporation of the resulting topology. This can be achieved by using the approach of Zeng et al. through comparative measurements, calculations, and adapted simulations [59]. Including the resulting mechanical properties and the expected hardness after shot peening would improve the model considerably. Due to the easy-to-implement logic of this framework, it is possible to apply this model to further mechanical surface treatments. Uprising technologies that are currently heavily investigated such as laser shock peening could be considered. A comparison of the three-dimensional FEA carried out by Li et al., also using the JC model to the two-dimensional model, will be considered [60]. Recent work from Dong et al. describes the development of a FEA for machining operations [61]. In this work, the effect on residual (tensile) stresses combined with a bimodal Gaussian function is used to predict existing stresses after machining and before mechanical surface treatment. This approach can be used to integrate the initial stress state of components to be shot peened. As a result, the accuracy of the initial white-box model presented in this work can be increased. Based on this combination, the number of practical experiments for the calibration of the algorithm can be further reduced. Recent work from Bock et al. [62] can additionally serve as a basis for the training of a physical data-driven artificial neural network.

**Author Contributions:** Conceptualization, B.J.R.; data curation, B.J.R. and M.S. (Marcel Sorger); formal analysis, M.S. (Marcel Sorger); investigation, K.H. and A.S.-G.; methodology, B.J.R. and M.S. (Marcel Sorger); project administration, B.J.R.; resources, M.S. (Martin Stockinger); software, B.J.R. and M.S. (Marcel Sorger); supervision, M.S. (Martin Stockinger); validation, B.J.R.; visualization, B.J.R. and M.S. (Marcel Sorger); writing—original draft, B.J.R. and K.H.; writing—review and editing, A.S.-G., B.J.R., and K.H. All authors have read and agreed to the published version of the manuscript.

**Funding:** This research received no external funding.

**Data Availability Statement:** The data presented in this study are available on request from the corresponding author.

**Conflicts of Interest:** The authors declare no conflict of interest.

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
