# Peer review of "Machine Learning Driven Prediction of Residual Stresses for the Shot Peening Process Using a Finite Element Based Grey-Box Model Approach"

_jmmp, doi:10.3390/jmmp5020039_

Round 1
Reviewer 1 Report
This manuscript reports the development and evaluation of a grey-box model to predict residual stresses generated by shot peening and to determine the peening parameters for specified residual stress. The developed model is interesting and useful for engineering applications. One major drawback is the unnecessary length of the manuscript due to the explanation of some textbook knowledge and those concepts that are irrelevant or loosely connected. For further consideration of the manuscript, the following issues should be addressed.
1. The writing needs to be concise. Some verbose paragraphs (e.g. lines 168-174, lines 188-209) should be deleted, and some lengthy sentences (e.g. Lines 255-261) should be shortened. Textbook knowledge should not be elaborated in the research paper and some irrelevant or loosely connected concepts should be avoided.
2. It is unclear about how the grey-box model is validated. More description about the experimental measurements should be added.
3. In Figure 7 and Figure 8, some 2D data plots are needed for comparison between experiments and simulations.
4. It seems the machine learning specifically refers to the regression. However, the efficacy of the regression has not been demonstrated. For complicated residual stress distribution, the regression may not work with required accuracy. In addition, examples of data training should be provided.
5. The limitation of the 2D FEA model should be discussed.
6. A section of conclusions should be added.
Author Response
Thank you for your precise comments!
Please see the attachment.

Reviewer 2 Report
Very nice work, directed to models of machine learning applied to FEM of shot peening. Little things to say, perhaps some weaker points:
FEM models are a complex aspect in mechanical treatments as it was shown by Surface improvement of shafts by the deep ball-burnishing technique, Surface and Coatings Technology 206 (11-12), 2817-2824 a direct competitor of shot peening. This group published other Works, including residual stresses analysis, please make a section about how mechanical treatments are key today. Peening and burnishing (also known as Low plasticity burnishing) are in competition, Lambda research for instance use both. Rodriguez is a leading author there.
Make a conclusion statement including latest news about MDPI and other journals, about the technologies. For instance, give them as highligths, one point per each claim.
Some references are old. JC is always a good model.
general aspect of the work is like a report more than a scietific work.all changes must be accomplish
Round 2
Reviewer 1 Report
The authors have addressed some issues concerning the reviewer. However, the following issues still require further clarification.
1. In Figures 6-8, were the experimental measurements conducted by the authors? If so, more details about the measurements should be added.
2. In Figures 6-8, why some of the residual stresses on the surface (depth close to zero) are tensile?
Reviewer 2 Report
Paper is OK
There are more references about ball burnishing and shaft rolling by some groups cited by you. You can provide the last version with the 2-3 missed one
